| Open Peer Review | Mycology | Methods and Protocols

# An experimental approach to investigate extracellular vesicle-mediated transfer of lipids between fungal cells

Lucas de O. Las-Casas,[1] Daniel A. Mellon,[1] Bárbara T. Bezerra,[1] Cássia M. Souza,[1] Flavia C. G. Reis,[1] Marcio L. Rodrigues[1,2]

**ABSTRACT**   Extracellular vesicles (EVs) are key mediators of communication in fungal populations, but the mechanisms underlying lipid exchange through EVs remain poorly characterized. Here, we describe an experimental approach to visualize lipid transfer between fungal cells using the lipophilic dye FM 1-43. This styryl dye efficiently stained membranous structures in multiple fungal species, including *Cryptococcus neoformans*, *Candida albicans*, *Aspergillus fumigatus*, *Saccharomyces cerevisiae,* and *Sporothrix brasiliensis*. Staining of the acapsular *C. neoformans* strain Cap67 with FM 1-43 generated fluorescent EVs, as shown by nano flow cytometry. Supernatants from FM 1-43-stained Cap67 cultures successfully transferred fluorescence to previously unstained cells, suggesting EV-mediated lipid exchange. The removal of EVs through ultracentrifugation of the supernatants effectively nullified this observation. A transwell system separating stained donor and unstained acceptor cells by a 0.4 µm membrane demonstrated that fluorescence transfer occurred without direct cell contact, with stronger signals observed when acapsular cells were used as donors. Using this model and acapsular *C. neoformans* cells as lipid acceptors, interspecies lipid transfer was detected when *Cryptococcus deuterogattii* and *C. albicans* were used as donor cells, although fluorescence levels were lower compared to intraspecies exchanges. Together, these results establish a tractable framework to monitor EV-mediated lipid trafficking among fungi. This model offers new experimental opportunities to dissect fungal communication mediated by EVs.

**IMPORTANCE**  Fungal cells release extracellular vesicles (EVs) that mediate intercellular communication, but the mechanisms and biological consequences of this process remain underexplored. Here, we provide experimental evidence that lipids can be exchanged between fungal cells via EVs, visualized using the lipophilic dye FM 1-43. This approach allows tracking of lipid trafficking between cells of the same or different species, even in the absence of direct contact. Understanding EV-mediated lipid exchange may offer insights into fungal physiology, signaling, and adaptation. These findings establish a model that can be broadly applied to study vesicle biology and intercellular communication in other microorganisms.

**KEYWORDS**   extracellular vesicles, *Cryptococcus*, cell biology

The urgent need for novel tools to combat fungal diseases, which annually claim millions of human lives (1), requires the characterization of cellular targets for the development of prophylactic, therapeutic, and diagnostic interventions. The cell wall and plasma membrane have been extensively studied in this regard (2). Recently, fungal extracellular vesicles (EVs) have emerged as structures of interest for the development of novel antifungal vaccines, therapeutic components, and diagnostic prototypes (3).

Fungal EVs are membranous compartments that are exported extracellularly by all species and morphological stages of fungi, to the best of our knowledge, studied so far

**Peer Reviewer** Nivea Pereira de Sa, Stony Brook University, Stony Brook, New York, USA

Address correspondence to Marcio L. Rodrigues, marcio.rodrigues@fiocruz.br.

The authors declare no conflict of interest.

See the funding table on p. 13.

(3). Historically studied as vehicles of secretion (4), it is now evident that in fungi, EVs serve as crucial mediators of various biological events at the population level, including the transfer of virulence (5), prion transmission (6), biofilm formation (7), morphological transition (8), and antifungal resistance (9, 10). Despite the clear role of fungal EVs in intercellular communication, there are no studies demonstrating, for instance, their cellular fate after transfer between fungal cells. These major gaps in the field are directly associated with the absence of experimental approaches to address them. In fact, a significant challenge in the field of fungal EVs lies in the limited availability of protocols to address major questions, such as their biogenesis, identification of biomarkers, understanding of their traversal of the cell wall, and their diversity (3).

Fei Mao (FM) dyes, small amphiphilic molecules, are extensively utilized as plasma membrane and vesicle trafficking dyes in fungi (11). These dyes are incapable of permeating biological membranes but reversibly associate with the outer leaflet of the lipid bilayer (12). FM dyes exhibit specificity for lipid bilayers involved in endocytosis and exocytosis (13). These dyes exhibit non-fluorescent properties in water but become intensely fluorescent upon membrane integration (14).

In this study, we developed and validated an experimental system to directly visualize and quantify lipid exchange mediated by EVs in the *Cryptococcus* model. Using the lipophilic dye FM 1-43, we demonstrated that live fungal cells release fluorescent EVs capable of transferring membrane lipids to unstained cells. Through a transwell model that separates donor and acceptor populations, we confirmed that lipid transfer occurs between living cells without physical contact with the participation of EVs. Extending this approach, we showed that interspecies lipid exchange also occurs, although less efficiently than intraspecies transfer. This model provides a platform to investigate the mechanisms and biological implications of EV-mediated communication in fungi.

## RESULTS

### Staining of fungal cells with FM 1-43

FM 1-43 is a lipophilic dye bearing a styryl ($C_6H_5CH=CH—$) group (15). This dye binds to membrane lipids in living cells and is commonly used to track endocytosis and exocytosis in several organisms (16). As already mentioned, these amphiphilic molecules cannot cross the lipid bilayer of biological membranes but reversibly associate with the outer leaflet. In water, they are non-fluorescent, but they become intensely fluorescent when integrated into the membrane (14). This dye has been successfully utilized for live-cell imaging of filamentous fungi (11). Therefore, we first tested the capacity of FM 1-43 to stain different species and forms of fungi at room temperature, including *Aspergillus fumigatus* (filamentous), *Candida albicans* (pseudohyphae), *Cryptococcus deuterogattii*, *Cryptococcus neoformans*, *Sporothrix brasiliensis*, and *Saccharomyces cerevisiae* (yeast cells; Fig. 1). Cellular borders were stained with calcofluor white (CFW), which binds to chitin in the cell wall, producing blue fluorescence. The structures stained with FM 1-43 (green fluorescence) had morphologies varying from intracellular, round-shaped, well-delimited organelles (*A. fumigatus*, *C. deuterogattii*, *C. neoformans*, *C. albicans,* and *S. brasiliensis*) to dispersed cytoplasmic structures (*S. cerevisiae* and *S. brasiliensis*). As a control, we stained an acapsular strain of *C. neoformans* at 4°C. In these cells, intracellular FM 1-43 staining was virtually absent, which agrees with an inhibition of endocytosis. In summary, for all species tested, FM 1-43 demonstrated the ability to efficiently stain membranous structures.

### An acapsular strain of *C. neoformans* stained with FM 1-43 produces fluorescent EVs

The efficient staining of intracellular structures of living fungal cells exclusively at room temperature suggested endocytosis, supporting the notion that lipid traffic remained active after FM 1-43 staining. In this context, we investigated whether secretory activity could be tracked with this dye, as suggested in previous studies utilizing FM 1-43 to

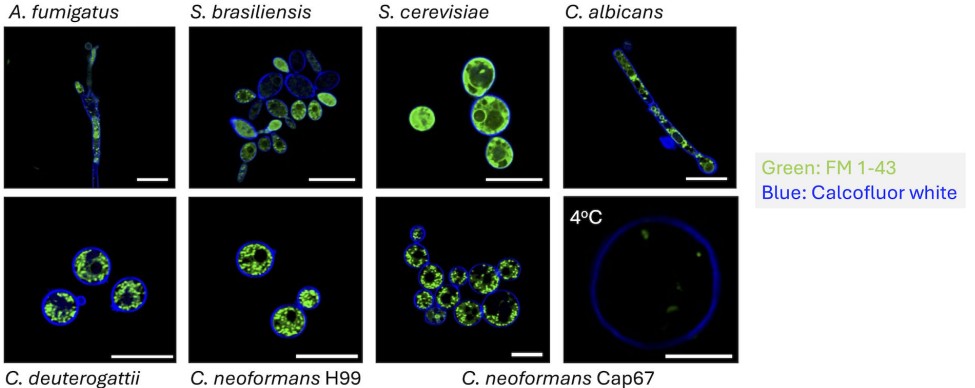

**FIG 1** Confocal microscopy of live fungal cells stained with FM 1-43 (intracellular green fluorescence) and CFW (blue fluorescence at the cell walls). All procedures were conducted at 30°C, resulting in intracellular dye incorporation and organelle staining. The only exception was the incubation of the acapsular strain Cap67 of *C. neoformans* at 4°C, where intracellular staining was virtually absent. Scale bars, 3 mm.

visualize synaptic vesicle exocytosis in the neuromuscular junction (16). For this assay, we initially selected the strain Cap67 of *C. neoformans*, based on our expertise with the *Cryptococcus* model and the absence of a capsule as a final barrier in the secretory process. These cells were stained with different concentrations of FM 1-43 for 30 min (Fig. 2A), washed with phosphate-buffered saline (PBS), and then incubated for 24 h in yeast peptone dextrose (YPD) medium. At this stage, the cultures were utilized for EV isolation and the analysis of vesicles through nano flow cytometry (17). EV analysis revealed the detection of fluorescent EVs in amounts that were directly correlated with the concentration of FM 1-43 used to stain the fungal cells (Fig. 2B). Cell-free medium incubated with similar concentrations of FM 1-43 was used as a control group to avoid the possibility that fluorescent nanoparticles corresponded to aggregates formed during the centrifugation steps (18). Under these conditions, detection of fluorescent nanoparticles was negligible (Fig. 2B). These results indicated that the Cap67 strain was able to produce FM 1-43-stained EVs. Due to the highest detection of fluorescent EVs, the FM 1-43 concentration of 100 µg/mL was selected for further experiments.

To determine whether the production of fluorescent EVs was exclusive to the Cap67 strain, we conducted a comparative experiment using the standard isolate H99 of *C. neoformans* after staining with FM 1-43. Fluorescent EVs were also detected in the supernatant of these cells (Fig. 2C), albeit in lower quantities than those observed with the acapsular strain. Notably, fluorescence intensity was higher in the EVs obtained from encapsulated cells (Fig. 2C). In the control group, no significant amounts of fluorescent particles were observed. Therefore, the presence of the polysaccharidic capsule might be a barrier to the export of EVs, but not a general inhibitor of their detection in cultures of FM 1-43-stained cells.

## Supernatants from Cap67 cells stained with FM 1-43 can transfer fluorescence to acceptor cells

The capacity of FM 1-43-stained cryptococci to generate fluorescent EVs and the literature reports indicating that fungal EVs can be transferred between distinct cells (5, 8, 19) prompted us to ask whether the transfer of fluorescent EVs can be monitored by confocal microscopy. For these experiments, we collected the supernatants from FM 1-43 stained cells (Cap67) for incubation with unstained acapsular cells for 1, 6, and 24 h, followed by analysis in the confocal microscope (Fig. 3). Alternatively, unstained cells were incubated with similar supernatants that underwent ultracentrifugation to deplete EVs. After 1 h of incubation, the levels of intracellular fluorescence were at the background levels in all systems. After 6 h, intracellular staining of organelles was clear in Cap67 cells incubated with the whole supernatant previously obtained from stained

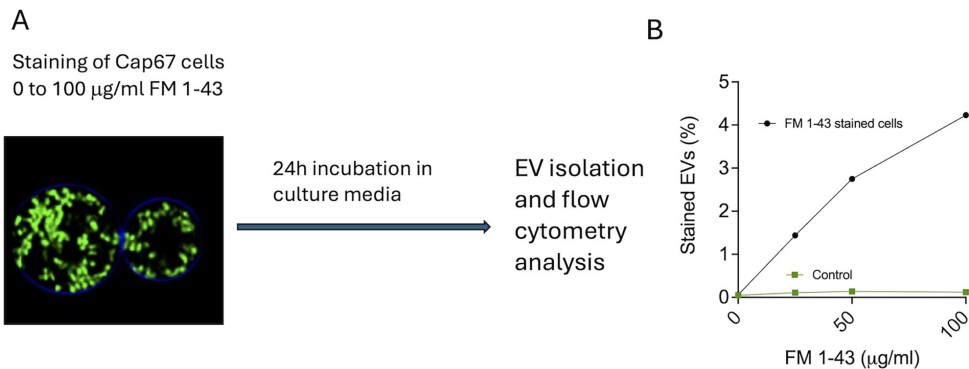

A

Staining of Cap67 cells
0 to 100 µg/ml FM 1-43

24h incubation in
culture media

EV isolation
and flow
cytometry
analysis

B

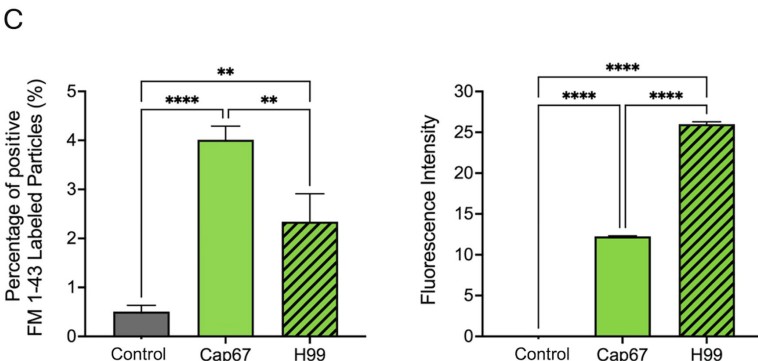

C

FIG 2 Detection of fluorescent EVs produced by FM 1-43-stained acapsular cells of *C. neoformans*. (A) Stained cells (Cap67) were incubated for 24 h in culture media, followed by EV isolation and analysis of the EVs by nano flow cytometry (B). While detection of fluorescent EVs increased as a consequence of dye concentration, the medium alone supplemented with FM 1-43 (control) produced negligible fluorescent levels. (C) Quantitative determination of fluorescent EVs produced by *C. neoformans* Cap67 and H99 cells stained with 100 µg/mL FM 1-43. The left panel represents the percentage of fluorescent EVs, while the right panel shows the intensity of EV labeling. While more fluorescent EVs were produced by Cap67 cells, the intensity of H99 EVs was higher. Data were analyzed using the ordinary one-way analysis of variance followed by Tukey's *post hoc* test for multiple comparisons (****$P$-value < 0.0001; **$P$-value < 0.05).

cells, in contrast to the EV-depleted supernatant, where fluorescence remained at background levels. Similar, but less intense fluorescent signals were observed after 24 h. These results suggest that EV-related fluorescence can be incorporated by unstained Cap67 cells.

## Live Cap67 cells stained with FM 1-43 cells transfer fluorescence to unstained acceptor cells in a transwell model

After confirming the capacity of EVs produced by stained Cap67 cells to transfer fluorescence to unstained acceptor cells, we investigated whether similar results would be observed using live cells. We then established a transwell model where FM 1-43-stained donor cells were placed in the upper compartment, and unstained Cap67 cells were placed in the lower transwell compartment. Acapsular cells were always used as acceptors to avoid any possible interference of the capsule in the process of lipid incorporation. In this transwell system, unstained and FM 1-43 stained cells were separated by polycarbonate membrane inserts with 0.4 µm nanopores that allow the passage of nanoparticles, such as EVs, but not entire cells (Fig. 4A). The systems were incubated at 37°C for 1, 6, and 24 h for further analysis of acceptor cells by confocal microscopy. Cap67 and H99 *C. neoformans* cells were utilized as donor cells (Fig. 4B).

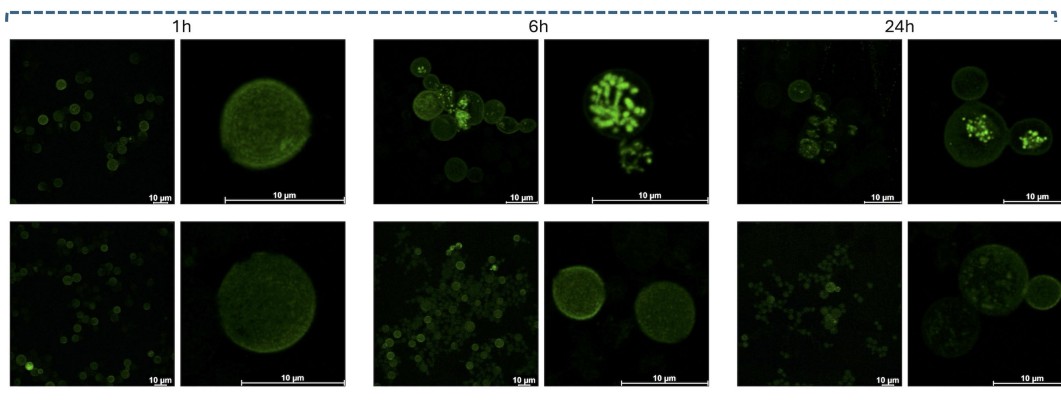

**Cap67 unstained cells plus supernatant from FM 1-43-stained cells**

1h 6h 24h

**Cap67 unstained cells plus ultracentrifuged supernatant from FM 1-43-stained cells**

**FIG 3** Incorporation of fluorescence by unstained Cap67 cells from supernatants obtained from FM 1-43-stained cells. The upper panels depict confocal analyses of unstained cells after incubation with the supernatants for 1, 6, and 24 h, demonstrating intracellular staining mostly after 6 h. For each incubation period, general (left) and amplified (right) microscopic fields are presented. The lower panels depict the microscopic profile of unstained cells incubated with the supernatants after ultracentrifugation for EV depletion. No intracellular fluorescence was observed. Scale bars, 10 µm.

Control systems comprised sterile media containing FM 1-43 at a concentration of 100 µg/mL. After a 1-h incubation period, acceptor cells exhibited background fluorescence levels in all systems, which were also observed at 6 and 24 h in the controls. After 6 h, fluorescent cells resembling those directly stained with FM 1-43 (Fig. 1) were observed in systems employing H99 or Cap67 isolates as donor cells (Fig. 4B). Notably, significantly higher fluorescence signals were observed when Cap67 cells served as the donors. These findings became even more evident after 24 h, thereby confirming the enhanced efficacy of Cap67 as donor cells. Microscopic observations in Fig. 4B were corroborated quantitatively by flow cytometry, as depicted in Fig. 4C. Collectively, these findings propose a model for investigating the mechanism of lipid transfer mediated by EVs in *C. neoformans*.

## Lipid transfer occurs between different fungal species, but it is more efficient within the *C. neoformans* donor-acceptor pairs

Until this point, we have demonstrated the capacity of different isolates to export lipids to Cap67 cells, exclusively within the *C. neoformans* model. However, recent findings support the existence of interspecies communication via fungal EVs (20). On another edge, the impact of secretion-oriented gene deletions on EV transfer in fungi remains unknown. In this context, we extended our transwell model (Fig. 5A) to encompass interspecies lipid transfer (*C. deuterogattii* versus *C. neoformans* Cap67; *C. albicans* versus *C. neoformans* Cap67) and the comparison between wild-type (WT) and mutant cells lacking the expression of genes associated with membrane architecture. For the latter, we compared WT *C. neoformans* with a mutant lacking expression of the Apt1 flippase and WT *C. deuterogattii* with a mutant where the scramblase Aim25 was deleted. Both enzymes were implicated in EV formation and membrane architecture in previous studies (17, 21–23). Confocal images revealed that H99 and *apt1△* cells efficiently transferred fluorescence to Cap67 cells after a 6-h incubation period, with no discernible differences in the staining patterns of the acceptor cells. Lipid transfer also occurred between *C. deuterogattii* (WT and *aim25△* cells) and *C. neoformans* Cap67 cells after 6 h of incubation, but the fluorescence signals were significantly more discrete. No discernible differences were observed between WT and *aim25△* as donor cells. These results were comparable to those obtained when *C. albicans* was used as donor cells, with even weaker fluorescence signals. The microscopical findings described in Fig. 5A were essentially confirmed by flow cytometry (Fig. 5B), with the exception of the *C.*

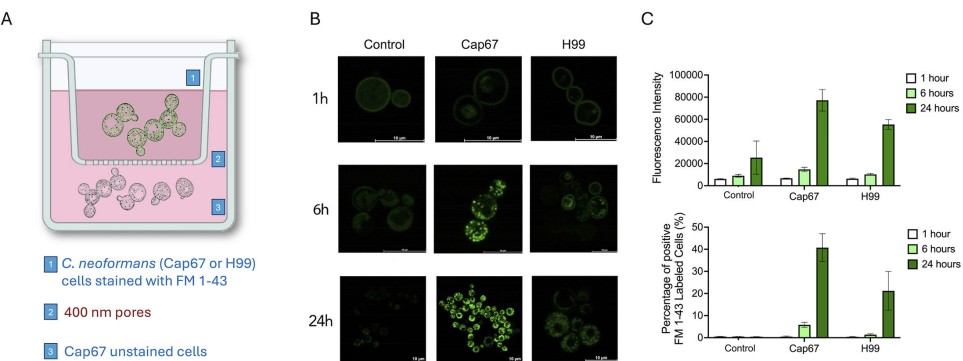

**FIG 4** Lipid transfer between *C. neoformans* cells in a transwell system. (A) Schematic representation of the transwell setup used to assess lipid transfer. FM 1-43-stained donor cells were placed in the upper compartment (1), while unstained acapsular Cap67 acceptor cells were placed in the lower compartment (3). The two populations were separated by a 0.4 µm polycarbonate membrane that allows the passage of EVs but prevents cell contact (2). (B) Confocal microscopy images of acceptor Cap67 cells after 1, 6, and 24 h of incubation with donor cells (H99 or Cap67) previously stained with FM 1-43. Fluorescent signal increased over time, with stronger staining observed when Cap67 cells served as donors. Scale bars, 10 µm. (C) Quantification of fluorescence intensity in acceptor cells by flow cytometry confirms the time-dependent and donor-dependent transfer of FM 1-43 fluorescence. The upper panel in panel C portrays relative fluorescence intensity, while the lower panel quantifies the percentage of fluorescent cells for each donor strain.

*deuterogattii* (WT and *aim25△* cells) pair. In this system, flow cytometry revealed a more efficient lipid transfer when mutant cells were used. These findings suggest interspecies lipid transfer, albeit at a reduced efficiency compared to the intraspecies model tested.

## Lipid incorporation by *C. neoformans* is an active process that occurs at different temperatures

The fact that direct intracellular staining was ineffective at 4°C (Fig. 1) indicates that lipid transfer is not a passive process, likely requiring endocytic mechanisms. To further explore this, we used our transwell model of lipid transfer with donor cells (H99) and acceptor cells (Cap67) at 30°C or 37°C. In this experiment, we included heat-killed Cap67 cells as acceptors. Lipid incorporation was observed at both temperatures after 6 h of incubation (Fig. 6), but only with living cells. While fluorescence signals were abundant in both living and heat-killed yeast cells at both temperatures after 24 h, the patterns of staining in dead cells were highly diffuse, contrasting with those in living yeast. These results suggest that lipid transfer in *Cryptococcus* is an active process that occurs at 30°C and 37°C.

## High-resolution imaging of lipid uptake by acceptor cells

Having validated the functionality of our model across multiple experimental conditions, most previous analyses focused on control validation, inter-strain comparisons, and population-level measurements. The intracellular fate of incorporated molecules, therefore, remained unclear. Identifying organelle-level localization was beyond the scope of this study, and earlier experiments did not include high-resolution images generated using maximum-projection mode. This approach merges all confocal fluorescence stacks, enhancing signal intensity. Although such enhancement can limit direct quantitative comparisons across systems, it yields highly detailed images that facilitate intracellular visualization. In our transwell model, FM 1-43-stained H99 or KN99 cells served as donors, and live Cap67 cells served as acceptors. High-resolution analysis (Fig. 7) revealed that lipid-derived fluorescence in acceptor cells was organized into structures whose size and morphology were consistent with EVs. These findings support a model in which donor cells release EVs that are subsequently internalized by acceptor cells and remain structurally intact within their intracellular environment.

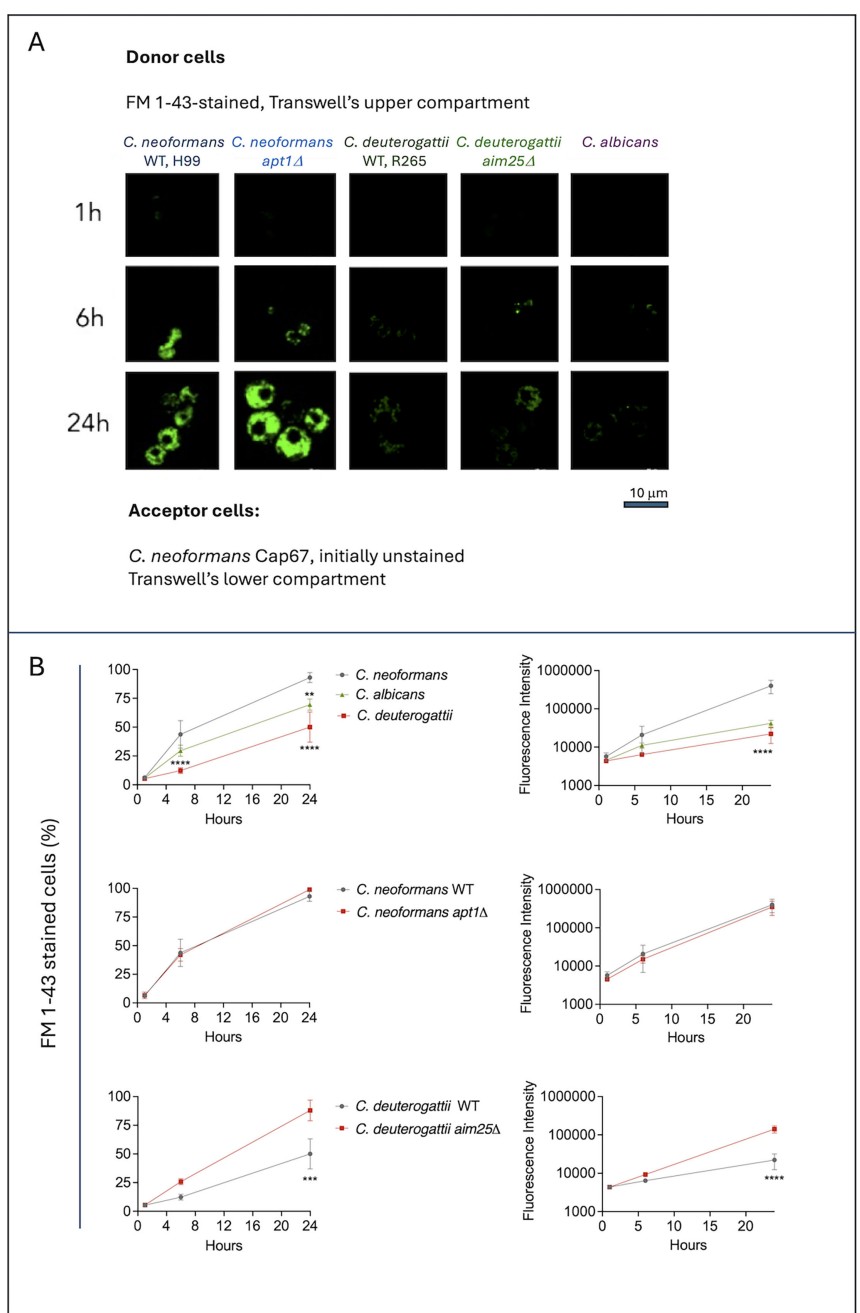

**FIG 5** Interspecies and interstrain analysis of lipid transfer. (A) Confocal microscopy images showing fluorescence transfer to acapsular *C. neoformans* Cap67 acceptor cells after 1, 6, and 24 h of incubation in the transwell system with different donor strains stained with FM 1-43FX. Donors included WT cells (*C. neoformans* or *C. deuterogattii*) and mutants of *C. neoformans* (*apt1Δ*) and *C. deuterogattii* (*aim25Δ*), as well as *C. albicans*. Fluorescence was detected in all donor-acceptor combinations, with stronger signals observed in the *C. neoformans* pairings compared to interspecies systems. (B) Flow cytometry quantification of the percentage of fluorescent cells (left panels) and fluorescence intensity (right panels) in acceptor Cap67 cells confirms that lipid transfer occurs across fungal species but is most efficient within *C. neoformans*. Data were analyzed using analysis of variance (ANOVA) followed by Tukey's *post hoc* test for multiple comparisons for *C. neoformans*, *C. albicans,* and *C. deuterogattii* comparisons (****$P$-value < 0.0001; **$P$-value < 0.05) and Šídák's multiple comparisons test for *C. neoformans* WT and *apt1Δ* strain comparisons, and *C. deuterogattii* and *aim25Δ* strain comparisons (****$P$-value < 0.0001; **$P$-value < 0.05). No significant differences were observed between WT and *apt1Δ* donor strains. These results

Fig 5 (Continued)

contrast with the comparison between WT *C. deuterogattii* and the *apt1△* strain, where mutant cells produced stronger staining. Data were analyzed using ANOVA followed by Šídák's multiple comparisons test (****P-value < 0.0001; ***P-value < 0.001).

## DISCUSSION

Fungal EVs serve as mediators of cell-to-cell communication processes (19, 24). Early studies on the yeast prion Sup35 have revealed that protein-based epigenetic information can be transmitted not only through direct cell contact or inheritance but also via yeast EVs (25). These vesicles are taken up by recipient yeast cells, where they induce self-sustaining aggregation of Sup35, effectively transferring prion-like states between cells (25, 26). This demonstrates that EVs can act as vehicles for the horizontal transmission of protein-based information, highlighting a mechanism of cell-to-cell communication that extends beyond traditional mechanisms of signaling. In *C. deuterogattii*, fungal populations can coordinate complex virulence behaviors through long-distance cell-to-cell communication mediated by EVs (5). In this model, virulence was enhanced by a "division of labor" mechanism. This cooperative behavior is regulated by EVs released by virulent strains, which can be internalized by host macrophages and trafficked to the phagosome, where they stimulate intracellular proliferation in less virulent cells. These findings underscore the significant role of EVs as mediators of intercellular signaling within fungal communities. EVs also possess the ability to transmit cues that modulate pathogenic potential and collective adaptation within the host. Notably, cryptococcal EVs have been demonstrated to modulate macrophage physiology (27). Concurrently, these observations reveal an active role of EVs in the communication of fungi with other fungal cells, as well as with the host. Other reports corroborate this perspective, indicating that EV-mediated communication regulates biofilm formation (28), antifungal resistance (10, 29), and morphological transition (8). Notably, a common limitation among the aforementioned studies is the identification of molecular regulators and the fate of EV components within cells that incorporate these structures. This highlights the urgent need for novel methodologies to elucidate the functions of fungal EVs, as extensively discussed in the recent literature (3, 30).

Live-cell imaging techniques have emerged as a promising new perspective in the study of fungal cell biology, enabling the analysis of organelle and molecular dynamics at high spatial resolution (31). One limitation of live-cell imaging in the study of fungal EVs is their reduced dimensions and lack of compartmentalization after extracellular

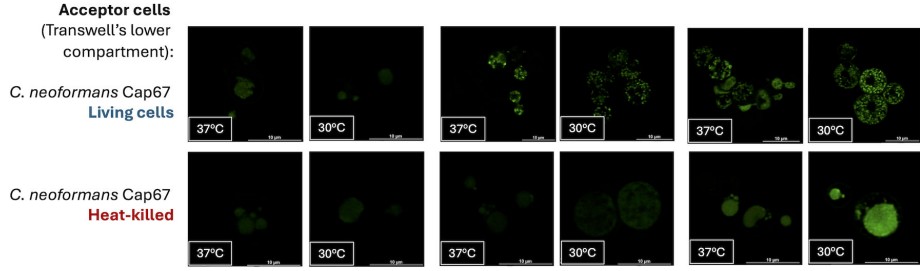

**FIG 6** Confocal imaging of living and heat-killed acapsular cells after incubation with WT donor cells. In the transwell model, FM 1-43-stained H99 cells were utilized as donor cells in experiments conducted at 30°C or 37°C. As acceptor cells (lower compartment of the transwell system), living or heat-killed acapsular (Cap67) yeast were employed. Intracellular staining was observed under all conditions, with fluorescence levels exhibiting variability and being higher in living cells. After 24 h, dead cells displayed abundant fluorescence, although their diffuse pattern contrasted with the well-punctuated pattern observed with living cells. Scale bars, 10 µm.

release. However, in the intracellular environment, vesicular structures can be visualized using fluorescence-based microscopic methods (27, 32). In our study, we observed that living fungal species can internalize the FM 1-43 dye, resulting in intracellular organelle staining due to dye endocytosis. This observation suggests that FM 1-43 traffic is efficient in fungi, and as previously described (11), it can occur in both endocytic and exocytic directions. This raises the possibility that EV transfer can be monitored in FM 1-43-stained cells. In fact, our results indicate that living, stained cells can produce fluorescent EVs, and supernatants from stained cryptococci serve as a source of fluorescent lipids that can be incorporated by acceptor cells. This process was EV-dependent, as depletion of EVs from the supernatants resulted in no incorporation of fluorescence by acceptor cells. In addition, high-resolution confocal microscopy revealed fluorescent structures compatible with EVs in acceptor cells. These results provided the basis for our contactless assay of cell-to-cell transfer of EVs, including intra- and interspecies models of lipid transfer.

Our transwell model provided direct evidence that lipid exchange between *C. neoformans* cells can occur without physical contact. The use of FM 1-43 allowed visualization and quantification of lipid transfer from stained donor to unstained acceptor cells, with significantly stronger fluorescence detected when the acapsular Cap67 strain acted as the donor. This suggests that the polysaccharide capsule, a hallmark of *C. neoformans*, may partially restrict vesicle release or lipid accessibility. The combination of confocal and flow cytometry analyses confirmed that EVs carry lipid components that can be functionally incorporated into recipient cells, offering a tractable model to dissect lipid trafficking dynamics in fungi. These findings extend current knowledge of fungal EV biology, revealing a measurable pathway for lipid redistribution that may influence cell physiology.

Building upon this system, we demonstrated that EV-mediated lipid transfer is not restricted to intraspecies interactions but can also occur between distinct fungal species, including *C. deuterogattii* and *C. albicans*. Although lipid transfer in these interspecies exchanges was markedly less efficient, the detection of transferred lipids indicates a conserved mechanism of EV-mediated communication across phylogenetically distant fungi. Mutants lacking key membrane-related enzymes, such as the flippase Apt1 and the scramblase Aim25, displayed no impairment in lipid transfer efficiency or moderate effects, suggesting that these proteins, while participating in EV biogenesis, are not individually required for lipid delivery. In any case, these results validate the currently described model for comparative analyses of EV transfer involving WT cells and secretory

**Donor cells** (Transwell's upper compartment):

FM 1-43-stained *C. neoformans* Cap67          FM 1-43-stained *C. neoformans* KN99

Acceptor cells
(Transwell's lower
compartment):

*C. neoformans* Cap67

**FIG 7**  High-resolution imaging of acceptor cells uptaking fluorescent lipids from different donor strains of *C. neoformans*. In the transwell model, FM 1-43-stained WT cells (strains H99, A; KN99, B) were utilized as donor cells in experiments conducted at 37°C. As acceptor cells, living Cap67 yeast were employed. Images were prepared under the maximum projection mode, which combines all fluorescence stacks in the confocal microscope, resulting in maximized fluorescent signals.

mutants, and possibly the interference of externally added molecules in EV-mediated communication.

In conclusion, our study establishes a versatile and quantitative framework for investigating EV-mediated lipid exchange in fungi. By combining fluorescence-based imaging, transwell assays, and flow cytometry, we demonstrated that fungal EVs can transport lipids both within and between species, independent of direct cell contact. These findings uncover a previously uncharacterized dimension of fungal communication that may contribute to collective adaptation, biofilm organization, and pathogenesis. Beyond *Cryptococcus*, the methodological approach described here can be applied to diverse fungal systems, offering new opportunities to explore the molecular machinery and physiological significance of EV-mediated interactions in microbial communities.

## MATERIALS AND METHODS

### Strains and culture conditions

The isolates utilized in our experiments encompassed the standard strains *C. neoformans* H99 and KN99, the acapsular strain Cap67 of *C. neoformans*, *C. deuterogattii* R265, *C. albicans* SC5314 (ATCC MYA-2876), *S. cerevisiae* W303-1a (ATCC 208353), *A. fumigatus* Ku80, *S. brasiliensis* 5110 (ATCC MYA-4823), and the mutant strains of *C. neoformans* (*apt1Δ*) (22) and *C. deuterogattii* (*aim25Δ*) (17). The latter strains are mutants lacking the expression of genes involved in EV production and cell membrane architecture (17, 21–23). The *Cryptococcus* and *Candida* isolates were maintained on Sabouraud dextrose agar plates (1% yeast extract, 2% peptone, 4% dextrose, and 1.5% agar), grown at 30°C for 24 h, and stored at 4°C. The *Saccharomyces* strain was maintained on YPD agar plates (1% yeast extract, 2% peptone, 2% dextrose, and 1.5% agar) and grown at 30°C for 24 h. Twenty-four hours prior to the experiments, the aforementioned strains were transferred to liquid YPD medium (1% yeast extract, 2% peptone, and 2% dextrose) and incubated at 30°C for 24 h with shaking (200 rpm). *A. fumigatus* was cultured on Sabouraud agar plates (1% [wt/vol] peptone, 4% [wt/vol] dextrose, and 1.5% [wt/vol] agar) at 35°C for 72 h. Conidia were scraped and suspended in 3 mL of sterile water containing 0.1% Tween 20. The suspension was homogenized for 15 s using a vortex mixer. The inoculum was utilized when the hyphae were less than 5%. Conidia were counted using a Neubauer chamber for the preparation of the final inoculum. *Sporothrix* spp. were cultured in Brain Heart Infusion broth (Difco) at pH 7.8 for 7 days under agitation (200 rpm) in Erlenmeyer flasks at 37°C. Cultures between the second and fifth passages were used. The suspensions were examined under a light microscope to ensure that the presence of hyphae was below 5%.

### Fungal staining

For staining of the cell wall, 50 µL of fungal cell suspensions were incubated with a CFW solution prepared at 5 µM (Sigma-Aldrich) in PBS for 30 min at room temperature, followed by washing (22). The lypophilic styryl dyes, FM 1-43 (unfixable) or FM 1-43*FX* (fixable) (Thermo-Fisher Scientific), were used to stain lipids present in cellular membranes (green fluorescence). Notably, the fixable version of the dye was exclusively employed for generating the results presented in Fig. 5 and associated text, owing to its enhanced fluorescence signals. To assess the ability of FM 1-43 to stain distinct fungal species, we modified a previously established endocytosis protocol suitable for live-cell microscopy (14). For this protocol, 50 µL of fungal cell suspensions were supplemented with FM 1-43 to a final concentration of 5 µg/mL in PBS. These suspensions were incubated for 30 min at room temperature, shielded from light, and manually shaken for 10 s every 10 min of incubation. Subsequently, the cells were washed three times and stored at 4°C. In the systems depicted in Fig. 5, fungal cells were stained similarly with FM 1-43FX. The cells were further analyzed by confocal microscopy.

## Preparation of extracellular fractions

For preparing stained EVs, fungal cells ($10^7$ cells) were suspended in 100 µL of FM 1-43 at 25, 50, and 100 µg/mL in PBS and incubated for 30 min at room temperature, shielded from light, and manually shaken for 10 s every 10 min of incubation. After washing in PBS, the cells were transferred to 10 mL of YPD and incubated at 37°C with shaking (200 rpm) for 1, 6, and 24 h (adapted from reference 14). Then, the cell suspensions were centrifuged at $5,000 \times g$ for 15 min at 4°C to remove cells. The resulting supernatants were collected and centrifuged at $15,000 \times g$ for 15 min at 4°C to remove debris (adapted from reference 17). The EVs were further analyzed by flow cytometry.

To prepare the supernatants containing FM 1-43-stained EVs or depleted of EVs, $10^8$ fungal cells were suspended in 100 µL of FM 1-43 (100 µg/mL in PBS). The cells were then incubated for 30 min at room temperature, shielded from light, and manually shaken by vortexing for 10 s every 10 min of incubation. Fungal cells were removed by centrifugation at $5,000 \times g$ for 15 min at 4°C. Then, debris was removed by centrifugation at $15,000 \times g$ for 15 min at 4°C (adapted from reference 17). The remaining supernatant was filtered through 0.45 µm syringe filters and collected for further assays as a source of stained EVs. Alternatively, the remaining supernatant was ultracentrifuged for 2 h at $100,000 \times g$ for depletion of EVs.

## Incorporation of fluorescent lipids by unstained Cap67 cells from culture supernatants

For lipid incorporation from the supernatants of FM 1-43-stained cells, $1 \times 10^8$ cells/mL of unstained Cap67 cells were suspended in 1 mL of YPD. This inoculum was then added to 9 mL of each of the supernatants obtained from FM 1-43-stained cultures (containing EVs or ultracentrifuged for 2 h) in a 250 mL Erlenmeyer flask. The flasks were incubated for 1, 6, and 24 h at 37°C with shaking (200 rpm) and protected from light. For each time point, 3.3 mL of the inoculum was collected. Each inoculum was immediately centrifuged at $2,600 \times g$ for 3 min, washed twice, resuspended in PBS, and stored at 4°C for further confocal analysis.

## Transwell assay

For the transwell assays of lipid transfer, $2 \times 10^7$ cells of FM 1-43-stained donor cells (200 µL in YPD) or medium alone supplemented with 100 µg/mL FM 1-43 (200 µL in YPD) were placed in the upper compartment of a polycarbonate transwell insert (Millipore Sigma-Aldrich) with a 0.4 µm pore-size membrane. In the lower compartment, $8 \times 10^5$ acceptor cells (Cap67) were added in 500 µL of YPD. The transwell system was incubated for 1, 6, and 24 h (37°C in a 5% $CO_2$ atmosphere or 30°C with no $CO_2$). Controls included heat-killed Cap67 cells ($2 \times 10^7$ cells incubated at 56°C for 30 min). At each time point, 500 µL of the acceptor cell suspension was collected from the lower compartment, washed three times with PBS, and stored at 4°C until analysis by confocal microscopy and flow cytometry.

## Flow cytometry

Fungal cells and EVs were analyzed using a CytoFLEX LX flow cytometer (Beckman Coulter). A FITC laser with a bandpass of 525/40 nm (green) was employed for FM 1-43 fluorescence detection. The blue side scatter (SSC, 488 nm) was utilized for cellular detection. The violet side scatter (VSSC, 405 nm) was employed for EVs, as it exhibits enhanced sensitivity for the detection of small particles (33). Cell populations were gated based on the SSC and FITC-A parameters. The gates were established using unstained Cap67 for cell-based experiments and unstained EVs for vesicle analysis. Data were analyzed using FlowJo software (version 10.0, BD Biosciences, USA), which permitted the quantification of median fluorescence intensity and the percentage of positively stained events.

## Confocal imaging

Confocal images of the isolates were captured using a Leica Stellaris 8 confocal laser scanning microscope (Leica Microsystems) at a 64× immersion objective. Blue fluorescence corresponding to CFW staining (cell wall) was acquired using a 4′,6-diamidino-2-phenylindole filter. Green fluorescence corresponding to FM 1-43 staining (membrane labeling) was acquired using a fluorescein isothiocyanate filter. Images were analyzed with LAS X software, using scale bars of either 3 or 10 µm. All samples were acquired under identical LAS X software settings for each condition. The pinhole was set to 1.0 Airy Units. Images were processed using Adobe Photoshop (version 26.4.1; Adobe Systems Inc.). Brightness adjustments were applied uniformly across entire images to enhance visualization, and no selective enhancement, removal, or alteration of any feature was performed. STELLARIS images were captured at a STELLARIS 8 system. Post‐processing was performed using the LAS-Stellaris or LIGHTNING modes. The LIGHTNING deconvolution approach reads out local image properties during image acquisition (pre-processing) and extracts suitable deconvolution parameters for the regularization procedure. Maximum-intensity projections were generated from the Z-stack data sets.

## Statistical analysis

Statistical analyses were conducted using GraphPad Prism software version 10.0 (GraphPad Software, Inc., La Jolla, USA). Group comparisons were subjected to one-way analysis of variance (ANOVA) or two-way ANOVA followed by Tukey's multiple comparison test or Šídák's multiple comparisons test. Statistical significance was determined when the $P$-values were less than 0.05.

## ACKNOWLEDGMENTS

M.L.R. was funded by the National Institute of Allergy and Infectious Diseases of the National Institutes of Health under Award Number R01AI183314, as well as CNPq grants 402651/2024-3, 404365/2023-0, and 304998/2022-2, and the Program for Research Stimulation (PEP) of the Carlos Chagas Institute of Fiocruz. This research was also co-funded by the UK Department of Health and Social Care (DHSC) as partof the Global AMR Innovation Fund (GAMRIF). This is a One Health UK aid fund that supports research and development around the world to reduce the threat of antimicrobial resistance(AMR) in humans, animals and the environment for the benefit of people in low- and middle-income countries (LMICs). The views expressed in this publication are those of the authors and not necessarily those of the UK DHSC. M.L.R. is on leave from a professor position at the Microbiology Institute of the Federal University of Rio de Janeiro. F.C.G.R. received a salary from the National Institute of Allergy and Infectious Diseases of the National Institutes of Health under Award Number R01AI183314. C.M.S. is a recipient of a fellowship from VPPCB/ Fundação Oswaldo Cruz. L.O.L.-C., D.A.M., and B.T.B. are graduate students at Programa de Pós-Graduação em Biologia Parasitária, Instituto Oswaldo Cruz, Fiocruz, Rio de Janeiro, Brazil.

The authors are grateful to the National Institute of Science and Technology in Human Pathogenic Fungi for their support. The authors thank theFlow Cytometry (RPT08L) and Confocal and Electron Microscopy (RPT07C) platforms of the Technological Platforms Network of the Oswaldo Cruz Foundation (Fiocruz).

## AUTHOR AFFILIATIONS

[1]Instituto Carlos Chagas, Fundação Oswaldo Cruz (Fiocruz), Curitiba, Brazil
[2]Instituto de Microbiologia Paulo de Góes (IMPG), Universidade Federal do Rio de Janeiro, Rio de Janeiro, Brazil

## AUTHOR ORCIDs

Marcio L. Rodrigues ⓘ http://orcid.org/0000-0002-6081-3439

## FUNDING

| Funder | Grant(s) | Author(s) |
|---|---|---|
| National Institutes of Health | R01AI183314 | Flavia C. G. Reis |
| | | Marcio L. Rodrigues |
| Conselho Nacional de Desenvolvimento Científico e Tecnológico | 402651/2024-3, 404365/2023-0, 304998/2022-2 | Marcio Rodrigues |
| Fundação Oswaldo Cruz | Program for Research Stimulation (PEP) | Marcio Rodrigues |
| Failsafe | FR1-49 | Marcio Rodrigues |

## AUTHOR CONTRIBUTIONS

Lucas de O. Las-Casas, Data curation, Formal analysis, Investigation, Methodology, Validation, Writing – original draft | Daniel A. Mellon, Investigation, Methodology, Writing – review and editing | Bárbara T. Bezerra, Investigation, Methodology, Writing – review and editing | Cássia M. Souza, Investigation, Methodology, Writing – review and editing | Flavia C. G. Reis, Investigation, Methodology, Writing – review and editing | Marcio L. Rodrigues, Conceptualization, Data curation, Formal analysis, Funding acquisition, Project administration, Resources, Supervision, Writing – original draft, Writing – review and editing

## ADDITIONAL FILES

The following material is available online.

### Open Peer Review

**PEER REVIEW HISTORY (review-history.pdf).** An accounting of the reviewer comments and feedback.

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
