## [Reviewer comments · Microbiology Spectrum]

Microbiology Spectrum

An experimental approach to investigate extracellular vesicle-mediated transfer of lipids between fungal cells

Lucas Las-Casas, Daniel Agreda-Mellon, Barbara Bezerra, Cassia Souza, Flavia Reis, and Marcio Rodrigues

Corresponding Author(s): Marcio Rodrigues, Instituto Carlos Chagas

Review Timeline:

Submission Date:	November 6, 2025
Editorial Decision:	November 27, 2025
Revision Received:	December 9, 2025
Accepted:	December 14, 2025

Editor: Agostinho Carvalho

Reviewer(s): Disclosure of reviewer identity is with reference to reviewer comments included in decision letter(s). The following individuals involved in review of your submission have agreed to reveal their identity: Nivea Pereira de Sa (Reviewer #2)

Transaction Report:

DOI: <https://doi.org/10.1128/spectrum.03604-25>

Re: Spectrum03604-25 (**An experimental approach to investigate extracellular vesicle-mediated transfer of lipids between fungal cells**)

Dear Dr. Marcio Rodrigues:

Dear Marcio,

Thank you for the privilege of reviewing your work. Below you will find my comments, instructions from the Spectrum editorial office, and the reviewer comments.

Revision Guidelines

Sincerely,
Agostinho Carvalho
Editor
Microbiology Spectrum

Reviewer #1 (Comments for the Author):

The authors demonstrate the ability to stain vesicle membrane with FM 1-43 dye in live cells and then demonstrate the ability of the labeled vesicles to be taken up by fungal cells within and across species using complementary confocal imaging and flow

cytometry. They further utilize a transwell assay show this does not require cell to cell contact. The method will be useful for tracking cellular vesicle release and uptake.

Couple of minor questions:

Did the authors confirm the secreted material was indeed vesicles?

Do the authors have higher resolution images to confirm the intracellular stained lipids are vesicles?

Similarly, higher resolution images may be useful to assess the intracellular fate of material that is taken up?

Is this an active process? Do dead cells take up vesicles?

Reviewer #2 (Comments for the Author):

The manuscript entitled "An experimental approach to investigate extracellular vesicle-mediated transfer of lipids between fungal cells" by Las-Casas et al. presents a well-designed and compelling experimental framework demonstrating that fungal extracellular vesicles (EVs) mediate lipid transfer between fungal cells, both within and between species. The authors employ the lipophilic fluorescent dye FM 1-43 in combination with nano-flow cytometry, confocal microscopy, and a transwell system to objectively visualize and quantify lipid exchange. This experimental model is straightforward, reproducible, and offers promising opportunities for future mechanistic studies on EV-mediated communication. The work clearly fills an important methodological gap in EV research and is highly relevant to fungal pathogenesis, cell-to-cell signaling, and antifungal resistance.

The manuscript is strengthened by its novel and impactful methodological approach, providing the first direct quantitative model to study EV-mediated lipid trafficking in fungi. The study convincingly demonstrates contact-independent lipid transfer as well as interspecies exchange, a significant contribution that broadens our understanding of fungal communication.

This work opens avenues for future studies aimed at determining the functional consequences of lipid transfer, including its relevance to fungal physiology, stress responses, biofilm formation, virulence, and antifungal drug tolerance. Experimental expansion in these directions would further increase the impact of this research.

The manuscript is clearly written and logically organized, with strong documentation and appropriate experimental controls. Results support the major conclusions. While minor adjustments are needed, the scientific content is solid.

Minor recommendations:

- 1) Correct typo on page 6, line 104: lypophilic dye - correction: lipophilic dye
- 2) Correct typo on line 636 and 678: Turkey's post hoc test- Tukey's post hoc test

Response to Reviewers

We thank the reviewers for their constructive comments. All changes made in the revised manuscript are highlighted in red, and new Figures 6 and 7 have been added as a result of our actions to address the pertinent points raised during review. Below, we provide a point-by-point response.

Reviewer 1

Reviewer's comments:

Did the authors confirm the secreted material was indeed vesicles?

Do the authors have higher resolution images to confirm the intracellular stained lipids are vesicles?

Similarly, higher resolution images may be useful to assess the intracellular fate of material that is taken up?

Authors' response and action taken during revision: Since these three points are interconnected, we have compiled a comprehensive response to them. In fact, these points are valid. Initially, evidence of EVs was obtained through low fluorescence signals after ultracentrifugation to eliminate EVs. However, we acknowledge that this experiment could be strengthened. Subsequently, we generated new high-resolution confocal images using the maximum-projection mode, which are now presented as Figure 7 and related text. These images reveal that the intracellular fluorescent structures in acceptor cells have sizes and morphologies that are compatible with EVs. A new section titled "High-resolution imaging of lipid uptake by acceptor cells" has been added to the Results to describe these findings. These images demonstrate that the fluorescent material remains as distinct vesicle-like structures rather than diffusing or integrating nonspecifically. The revised Results text clarifies that organellar-level localization was beyond the scope of this study, but the new imagery suggests that EVs maintain their structural integrity within the cell.

Reviewer's comment: Is this an active process? Do dead cells take up vesicles?

Authors' response and action taken during revision: To address this question, we performed new experiments now shown in Figure 6. Using the transwell model, we compared living and heat-killed acceptor cells. We observed:

- Active, punctate intracellular vesicle incorporation only in living cells,
- Diffuse, nonspecific fluorescence in heat-killed cells, especially at later time points.

These results indicate that EV-mediated lipid incorporation is predominantly an active, physiologically regulated process. A new paragraph was added to the Results (“Lipid incorporation by *C. neoformans* is an active process...”).

Reviewer 2

1) Reviewer’s comment:

Correct typo on page 6, line 104: *lypophilic dye* → *lipophilic dye*.

Authors' response and action taken during revision:

Corrected as requested.

2) Reviewer’s comment:

Correct typo on lines 636 and 678: *Turkey's post hoc test* → *Tukey's post hoc test*.

Authors' response and action taken during revision:

Corrected in all instances.

Re: Spectrum03604-25R1 (**An experimental approach to investigate extracellular vesicle-mediated transfer of lipids between fungal cells**)

Dear Dr. Marcio Rodrigues:

Dear Marcio,

Your manuscript has been accepted, and I am forwarding it to the ASM production staff for publication. Your paper will first be checked to make sure all elements meet the technical requirements. ASM staff will contact you if anything needs to be revised before copyediting and production can begin. Otherwise, you will be notified when your proofs are ready to be viewed.

Sincerely,
Agostinho Carvalho
Editor
Microbiology Spectrum

Reviewer #1 (Comments for the Author):

Queries addressed. Thanks